# Active Learning for Optimal Minimization of Experimental Characterization Uncertainty

**Marcus Schwarting**
Dept. of Computer Science
University of Chicago
5801 Ellis Avenue
Chicago, IL 60637
meschw04@uchicago.edu

**Nathan Seifert**
Dept. of Chemistry
University of New Haven
300 Boston Post Road
West Haven, CT 06516

**Logan Ward**
Dept. of Data Science and Learning
Argonne National Laboratory
9700 Cass Avenue
Lemont, IL 60439

**Ben Blaiszik**
Dept. of Data Science and Learning
Argonne National Laboratory
9700 Cass Avenue
Lemont, IL 60439

**Ian Foster**
Dept. of Computer Science
University of Chicago
5801 Ellis Avenue
Chicago, IL 60637

**Yuxin Chen**
Dept. of Computer Science
University of Chicago
5801 Ellis Avenue
Chicago, IL 60637

**Kirill Prozument**
Dept. of Physical Sciences
Argonne National Laboratory
9700 Cass Avenue
Lemont, IL 60439

## Abstract

Collecting experimental measurements is rarely an end in itself; rather, measurements inform key outcome statistics. Standard active learning procedures can drive a cumulative decrease in measurement uncertainty, but do not account for the uncertainty of the outcome. Here, we present an active learning framework that collects measurements agnostic to specific outcomes but which minimize outcome uncertainty, and demonstrate its applicability with imaging and spectroscopic tasks. We show how our framework can effectively select regions for measurement without iteratively retraining a model. We conclude with two instances where our framework has outperformed standard active learning procedures to accelerate the classification of unknown samples.

## 1 Introduction

Accurate characterization of an unknown chemical or material can be a time- and resource-intensive process, especially in a context-free setting. Characterizing a sample often requires obtaining multiple individual noisy measurements that, when combined, yield the desired identifying information. Effectively allocating resources to characterizing a sample is an important challenge with many applications. We present a novel active learning (AL) approach for negotiating the trade-off between time and precision that allows a researcher to characterize a sample reliably with fewer measurements. We start with a pre-trained supervised model with built-in uncertainty quantification (UQ), and iteratively select characterization routines to reduce model uncertainty optimally. While previous works select from a broad set of measurements to improve retrained model performance (and often do not repeat measurements), our approach accounts for even extremely noisy individual measurements to drive towards minimizing classification uncertainty with a fixed trained model.

38th Conference on Neural Information Processing Systems (NeurIPS 2024).

Spectroscopy is an important area of analytical chemistry where a typical workflow involves iteratively recording noisy measurements until a domain expert can draw conclusions about the contents of a sample. AL has been applied for certain spectroscopic sub-tasks [21], but has never before been applied to rotational spectroscopy. Rotational spectroscopy uses microwave and millimeter-wave frequencies (ca. 2–800 GHz) to probe transitions between the quantized energy levels of molecules associated with their overall angular momentum [35]. The observed rotational spectrum is extremely sensitive to molecular structure and intramolecular interactions. Rotational spectroscopy is unique in both the structural insights its spectra provide and the rich data it carries for machine learning (ML) applications [38]. Rotational spectra can be acquired via broadband or narrowband measurements. Broadband rotational spectroscopy [3, 33] has enabled researchers to acquire thousands of rotational transitions in a single measurement, with simultaneous acquisition of $\sim 10^5$ resolution elements at a typical signal-to-noise ratio (SNR) of $\sim 10^4$ [25, 36]. In broadband rotational spectroscopy, transitions from an entire spectral band can be acquired almost instantaneously, with repeated measurement averaging eventually leading to an acceptable SNR [37]. Alternatively, a full spectrum can be obtained from multiple sequential narrowband measurements, where a decrease in bandwidth corresponds to an increase in SNR at a rate of $(\Delta\nu)^{-\frac{1}{2}}$ [3]. For a fixed measurement SNR, there is no statistical advantage to either averaging broadband measurements or sequential narrowband measurements [23]. However, when an outcome is based on decreasing molecule identification uncertainty, rather than decreasing measurement uncertainty, we find that our AL framework can efficiently select narrowband measurements to reduce classification uncertainty.

## 2 Background

Methods for UQ can use an ensemble (Monte Carlo)-based approach or an incorporated Bayesian approach. For a classification task, a model with UQ will generate a probability distribution for each class, centered at the predicted class probability. Classical ML models such as random forest can give both a prediction and UQ derived from the ensemble. A close deep learning equivalent can be naively achieved by training an ensemble of feed-forward architectures, or via a dropout regularization approach. For an overview of ensemble deep learning approaches for UQ, see Gawlikowski et al. [13]. Alternatively, UQ can be a baked-in part of model training and inference, as in Gaussian process regression (or kriging) [28]. After setting a statistical prior based on previously collected data, a Gaussian process regressor makes calibrated predictions from inputs, including both a prediction mean and a standard deviation [11]. For more information on models that incorporate UQ as part of inference, see Abdar et al. [1].

Data acquisition using AL can be divided into approaches that iteratively retrain the prediction model and those that keep the prediction model fixed [31]. The former is often referred to as active instance labeling (e.g., [34]), and the latter as active feature evaluation (e.g., learning with feature cost [17], interactive troubleshooting [7] and medical diagnosis [2, 6]). When the model is iteratively updated, the objective is to select points to improve model performance [27]. When the model is fixed, active learning operates on a fixed pre-trained model where decisions must be made in serial to come to a conclusion with high confidence [5, 26].

Whether a UQ model is fixed or iteratively updated, selection of the acquisition function can significantly alter the progress of an AL approach. Consider a classification task with $N$ classes, and where $T$ samples are drawn iid from a Bayesian prior. For a fixed patch index $i \in \{1, ..., r\}$, we can establish probabilities $\boldsymbol{P}^{i,t} = [P_1^{i,t}, ..., P_N^{i,t}]; \ t \in \{1, ..., T\}$. Furthermore, for class $n$, let $\bar{P}_n^i = \frac{1}{T}\sum_{t=1}^{T} P_n^{i,t}$. The acquisition functions can be broken into information theoretic approaches (based on entropy) and standard deviation approaches [12]. We focus on four acquisition functions:

- Predicted information entropy [32], defined as $\text{ENTR}(\boldsymbol{P}^i) = -\sum_{n=1}^{N} \bar{P}_n^i \log \bar{P}_n^i$.

- Bayesian AL by disagreement, or BALD [15, 18]—akin to an expected entropy improvement metric. $\text{BALD}(\boldsymbol{P}^i) = \text{ENTR}(\boldsymbol{P}^i) + \frac{1}{T}\sum_{n=1}^{N}\sum_{t=1}^{T} P_n^{i,t} \log P_n^{i,t}$.

- Mean standard deviation [16], defined as $\text{MSTD}(\boldsymbol{P}^i) = \frac{1}{N}\sum_{n=1}^{N} \sqrt{\frac{1}{T}\sum_{t=1}^{T}(P_n^{i,t} - \bar{P}_n^i)^2}$.

- Baseline uniform acquisition, where $\text{UNIF}(\boldsymbol{P}^i)$ is drawn i.i.d. from $U(0, 1)$.

## 3 Methods

We start with a demonstration using the MNIST dataset of 28×28-pixel gray-scale images of handwritten digits [8]. We train a feed-forward neural network on MNIST with a standard 80/20 train/test split, augmenting training images with added Gaussian noise [30]. We break a 28×28 image from the test set into 4×4 image patches. For a ground-truth patch $x^i \in \mathbb{R}^{16}$, we can draw measurements from a "noisy oracle" as $\hat{x}^i = x^i + \varepsilon; \; \varepsilon \sim N(\mathbf{0}, \mathbf{\Sigma}_M)$. We use a multivariate Gaussian prior $\pi$ with mean $\mu$ and covariance $\mathbf{\Sigma}$ for each 4×4 patch from the MNIST training set (see Appendix B). Comparing central vs. perimeter patches suggests that patches with higher prior variance will be more useful to measure (satisfying a pure Bayesian exploration objective). But prior variance cannot be the only selection criterion, since measurements of patches with high prior variance need not reduce classification uncertainty.

Next we show a similar demonstration with rotational spectroscopy. We select the molecule propene $(CH_3CHCH_2)$ as our ground truth class for subsequent tests [20, 24]. We then create similar classes of hypothetical molecules with spectra resembling propene. Additional spectroscopic details, along with how these similar spectral classes were created and augmented for training robust classifiers, is given in Appendix A. We choose to simulate spectra in the 65–85 GHz range, which corresponds to a common experimental apparatus range [37]. Also, we bin and renormalize all spectra into either $b = 20$ or $b = 200$ equally sized patches. While a variety of supervised classification models could be readily employed, we opted for a logit classifier with a one-versus-rest scheme. Regardless of the model choice, the result is expressed as $\mathcal{M} : \mathbb{R}^b \to [P_1, ..., P_N]$. Finally, we use the mean vector and covariance matrix across all molecule spectra in the PC9 dataset of small organic molecules [14].

When querying a noisy oracle, we assume samples are drawn from a multivariate Gaussian likelihood with covariance $\mathbf{\Sigma}_M$. Therefore, we can use a conjugate prior/posterior that is also a multivariate Gaussian [10]. Algorithm 1 shows our acquisition workflow. For a given patch we sample from the prior, impute the sample, and run model inference. From the classification probabilities, we select a patch using the acquisition function $f \in \{BALD, MSTD, ENTR, UNIF\}$. After making a measurement, we update the prior. Appendix C shows how we can generalize to both a Poisson noise model or regression tasks. We benchmark acquisition function loss via a the remaining model probability entropy after a measurement is captured.

---

**Algorithm 1:** Active learning workflow for experimental acquisition across patches.

---

**Initialize:** Prior $\pi$ with mean $\mu$ and covariance $\mathbf{\Sigma}$, model $\mathcal{M}$, measurement covariance $\mathbf{\Sigma}_M$.

**for** $q = 1, ..., Q$ **do**                                             `// Measurement Loop`

    **for** $i = 1, ..., r$ **do**                              `// Patch Selection Loop`

        **for** $t = 1, ..., T$ **do**                       `// Monte Carlo Sample Loop`

             $\mathbf{I}^{i,t} \sim \pi_i(\cdot); \; \mu_i^* = [..., \mu^{i-1}, \mathbf{I}^{i,t}, \mu^{i+1}, ...]$ ;      `// Sample Imputation`

             $\mathbf{P}^{i,t} = [P_1^{i,t}, ..., P_N^{i,t}] = \mathcal{M}(\mu_i^*)$ ;      `// Inference on Imputed Sample`

    $s = \underset{i \in \{1,...,r\}}{\operatorname{argmax}} f(\mathbf{P}^i)$ ;             `// Patch Selection From Acquisition f`

    $\hat{\mathbf{I}}_s = \mathbf{I}_s + \varepsilon; \; \varepsilon \sim N(\mathbf{0}, \mathbf{\Sigma}_{M,s})$ ;       `// Measurement Capture on Patch s`

    $\mu_s \leftarrow (\mathbf{\Sigma}_s^{-1} + \mathbf{\Sigma}_{M,s}^{-1})^{-1}(\mathbf{\Sigma}_s^{-1}\mu_s + \mathbf{\Sigma}_{M,s}^{-1}\hat{\mathbf{I}}_s)$;

    $\mathbf{\Sigma}_s \leftarrow (\mathbf{\Sigma}_s^{-1} + \mathbf{\Sigma}_{M,s}^{-1})^{-1}$;                `// Prior Update on Patch s`

**return** $\mathcal{M}(\mu) = \mathbf{P}^{final}$ ;              `// Final Model Classification`

---

## 4 Results

Figure 1 (left) shows the outcome of 1000 guided measurements on patches of an MNIST test image, with error bounds constructed over 50 runs on the same test image. Insets show the posterior updates at intervals of 250 iterations for each acquisition function. For measurements using BALD and MSTD acquisition functions, a quantitative improvement in the loss function is reflected qualitatively when we see that the true digit (a "2") more discernible for BALD and MSTD than for UNIF or ENTR. The ENTR acquisition function quickly becomes stuck taking measurements at the periphery where prior measurement variances start off small and downstream classification uncertainties barely change, so

there is virtually no change in the measurement posterior. The UNIF acquisition captures superfluous measurements that have little impact on either the posterior or model scoring.

Figure 1 (right) shows the outcome of 1000 guided measurements on 20 patches of the rotational spectrum of propene with nine additional classes, with error bounds constructed over 50 runs on the same ground truth spectrum. While there is no statistical guarantee that either BALD or MSTD will always outperform UNIF, however using a one-sided t-test we can be confident that both BALD ($t = -7.17$, $p < 10^{-9}$) and MSTD ($t = -4.16$, $p < 10^{-5}$) are likely to outperform UNIF. Table 1 shows three further tests which vary the number of patches (either 20 or 200, representing broadband and narrowband modes) as well as the number of molecule classes (either 10 or 100). BALD and MSTD significantly outperform ENTR and UNIF on all four tests. While these acquisition methods are agnostic to the correct classification probability, Table 1 presents these probabilities to provide clarity and as a proxy for problem difficulty (which increases with number of classes and decreases with patch count). A higher correct classification probability does not perfectly correlate with a lower loss, since our loss is defined as the final model entropy after all measurements have been collected.

BALD and MSTD offer the best improvement for measurement acquisition when individual measurements are not so noisy that the posterior update changes only slightly at each iteration (such as the second row). If individual measurements are less noisy, then our framework yields a process approaching a standard AL model driven by an exploration objective. Furthermore, in less noisy settings, averaging broadband measurements is likely to be more effective for decreasing classification uncertainty than sequential narrowband measurements. When comparing broadband (20 patches) experiments versus narrowband experiments (200 patches), we see that the same number of measurements (with adjusted SNR) yields a much better loss for BALD and MSTD strategies. In other words, our AL framework can rapidly decrease molecule identification uncertainty when there are many patches to choose from.

## 5   Conclusion

In this work, we presented an active learning framework for measurement acquisition that works to decrease uncertainty in classification rather than measurement. We demonstrate how our framework can operate effectively in both an imaging and a spectroscopic context to drive towards accurate classification. We also demonstrate how our AL framework can effectively select from among many possible narrowband measurements to disambiguate between similar molecule species. In the

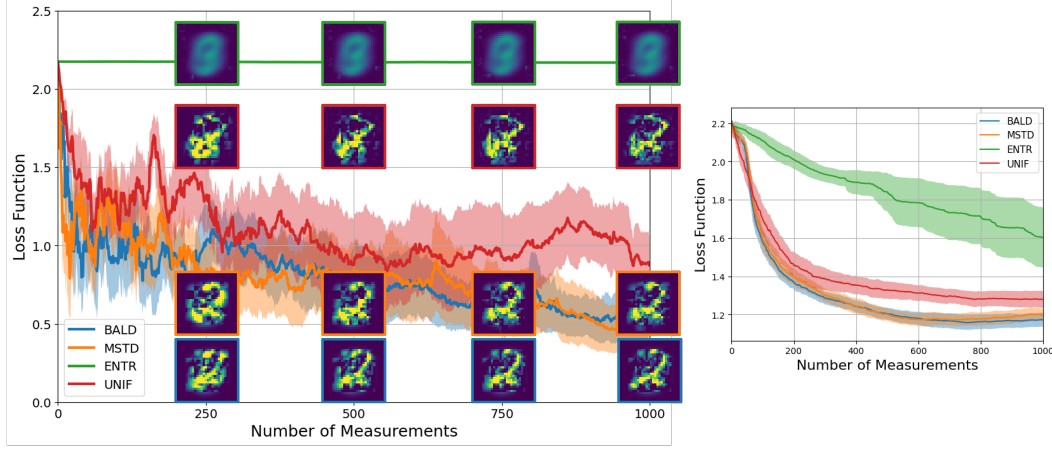

Figure 1: Left: Classifier loss function over 1000 MNIST sample measurements selected by using different acquisition functions (BALD, MSTD, ENTR, and UNIF). Inscribed images show the Bayesian posterior reconstruction using each acquisition function at intervals of 250 iterations (from left to right). Shaded regions indicate a bound of $\pm \frac{1}{2}\sigma$ for each acquisition function. Right: Loss function calculations across acquisition functions over 1000 rotational spectrum measurements over 20 bins with ten total classes. Tests are run over 50 iterations for each acquisition function, and filled-in regions indicate a bound of $\pm \frac{1}{2}\sigma$ for each acquisition function.

Table 1: Final loss across 1000 spectral measurements across acquisition functions, along with the final correct classification probability $(P_c)$. Logit model accuracy on train and test sets are also given.

| Patch Count | Class Count | Model Accuracy Train (Test) | BALD $(P_c)$ | MSTD $(P_c)$ | ENTR $(P_c)$ | UNIF $(P_c)$ |
|---|---|---|---|---|---|---|
| 20 | 10 | 0.964 (0.947) | **0.719 (0.817)** | 1.086 (0.712) | 1.867 (0.344) | 1.333 (0.568) |
| 20 | 100 | 0.849 (0.807) | 3.074 (0.183) | **3.032 (0.193)** | 3.787 (0.131) | 3.276 (0.140) |
| 200 | 10 | 0.993 (0.985) | 0.116 (0.983) | **0.090 (0.987)** | 1.915 (0.314) | 0.890 (0.804) |
| 200 | 100 | 0.899 (0.871) | 0.796 (0.845) | **0.668 (0.878)** | 2.156 (0.138) | 2.300 (0.402) |

future, we hope to generalize this approach to eventually characterize the sorts of complex mixtures present in an analytical chemistry context. We also hope to demonstrate the viability of our approach for regression and segmentation tasks, which may have applications in areas of microscopy and tomography.

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

## A  Details of Rotational Spectroscopy

Rotational spectroscopy is a powerful analytical technique, but faces hurdles to greater adoption in industry settings due to the expertise required to manually interpret spectra and the amount of time required to capture reliable spectra. To a first approximation, the rotational spectrum of a molecule (assuming a single low-energy structure) can be described with three inertial constants $A \geq B \geq C$, usually presented in units of MHz, that primarily control peak locations. There are also three corresponding dipole constants $\mu_A, \mu_B, \mu_C$, usually presented in units of Debye, that primarily control peak intensities. These six constants are a near-unique representation of the rotational spectrum of a low-energy molecular structure, at least to a first approximation [29]. There are three types of peaks present in a rotational spectrum: A-type peaks (which depend on $B$ and $C$ constants), B-type peaks (which depend on $A$ and $C$ constants) constants, and C-type peaks (which depend on $A$ and $B$ constants). The impressive SNR provided by rotational spectroscopy means that peak frequencies can be measured to <10 kHz precision (corresponding to around seven or eight significant figures). Peak intensities, on the other hand, can vary more widely due to several quantum and thermodynamic effects that are difficult to properly account for beforehand.

We chose propene as a reference species because is easily measured using standard microwave instruments [20], is limited to one low-energy conformer at low temperatures [9], and is of particular astrochemical interest [4]. Based on density functional theory calculations (performed using ORCA [22] with a B3LYP functional and a 6-31g(d) basis set), we can estimate the rotational constants of propene to be $A^* = 46753.53$ MHz, $B^* = 9239.72$ MHz, $C^* = 8101.64$ MHz. Likewise, we can estimate the dipole constants of propene to be $\mu_A^* = -0.343$ Debye, $\mu_B^* = 0.085$ Debye, $\mu_C^* = -0.001$ Debye. We then create similar classes of hypothetical molecules with very similar spectra to propene. After propene, we create either 9 or 99 additional classes with constants obtained by sampling $\chi_i \sim U(\chi^* - \delta, \chi^* + \delta)$; $\chi \in \{A, B, C\}$, setting $\delta$ to 100 MHz. This creates a set of class-specific spectra that are closer to the spectra of propene than those of other similar structures. For smaller $\delta$, classes will more closely resemble one another and the classification model will have a harder time distinguishing between classes. We augment spectral intensities by adding Gaussian noise in the frequency domain with a standard deviation of 0.1, with a floor set at zero (since intensities cannot be negative). We augment peak locations by randomly adjusting rotational constants according to $\chi_i^\dagger \sim N(\chi_i, \sigma^2)$; $\chi \in \{A, B, C\}$. This effectively adjusts A-type, B-type, and C-type peaks individually, and can help account for the known differences between simulated constants and constants identified experimentally [19]. We augment each class over 1000 random iterations, then train our logit classifier.

## B  Constructing Prior Distributions

We show the prior distributions for the MNIST training data in Figure 2 and for the rotational spectroscopy example in Figure 3. We use a Gaussian noise model to express the prior of both the MNIST and the rotational spectroscopy training set, binned into 100 MHz regions. For rotational spectroscopy, the acquisition of FID measurements is expected to inherently follow a Gaussian noise model (although not necessarily with nonzero covariances between bins).

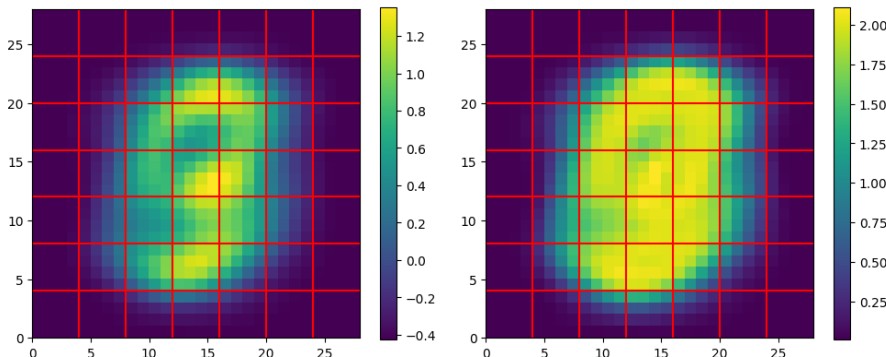

Figure 2: *Left*: Average image from all training images in MNIST dataset, delimited into 4×4 squares to show sample patches. *Right*: Variance image of all training images in MNIST dataset (with added extra variance of 0.01 for all patches), delimited into 4×4 squares to show sample patches.

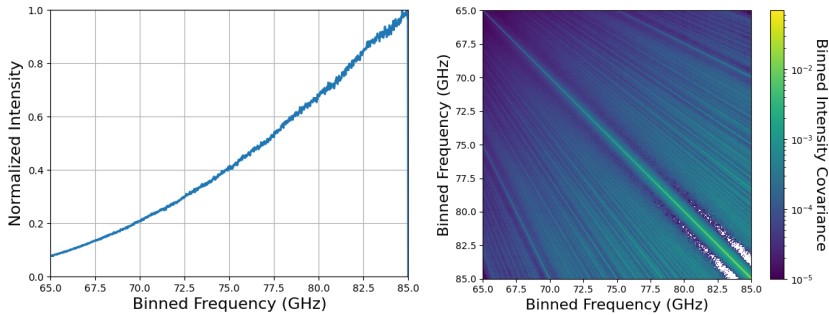

Figure 3: *Left*: Average normalized rotational spectrum of all molecules in PC9, set into 100 MHz bins. *Right*: Covariance matrix of normalized rotational spectra for all molecules in PC9, set into 100 MHz bins.

## C   Derivations for Additional Acquisition Tasks

In the main body of this work, we show how our active learning framework can operate given a Gaussian noise model and a classification objective. Here we suggest that our approach can also work with a Poisson noise model and for a regression objective. This requires adjusting two equations: the conjugate prior update (stated in Algorithm 1) and the acquisition function (stated in Section 2). Assuming we are sampling from a Poisson likelihood noise model, we rely on a conjugate prior and posterior that is Gamma-distributed. We must therefore construct an initial Gamma-distributed prior which, with respect to $\alpha_s$ and $\beta_s$, is updated with a new observation $\hat{I}_s$ as $\alpha_s \leftarrow \alpha_s + \hat{I}_s$ and $\beta_s \leftarrow \beta_s + 1$. Note that because there is no documented conjugate prior relationship for a multivariate expansion of a Poisson likelihood (that we know of), we must treat each index of our distributions as a distinct and independent Poisson random variable.

Next, suppose we wish to optimally decrease measurement uncertainty for a regression task. While our conjugate priors remain the same for new observations, our acquisition functions must change. Suppose our UQ model (such as a Gaussian process regressor) operates as $\mathcal{M} : \mathbb{R}^b \to (x_t, \sigma_t^2)$. Suppose for a set of Monte Carlo samples on a patch, we perform model inference to obtain $\{(x_1, \sigma_1^2), ..., (x_T, \sigma_T^2)\}$. First we could simply take $\text{MSTD}(\boldsymbol{x}, \boldsymbol{\sigma^2}) = \sqrt{\frac{1}{T} \sum_{t=1}^{T} (x_t - \bar{x})}$ where $\bar{x} = \sum_{t=1}^{T} x_t$, or the mean standard deviation of the output predictions (excluding predicted variances). We can also reformulate a predicted information entropy equation as $\text{ENTR}(\boldsymbol{x}, \boldsymbol{\sigma^2}) = \frac{1}{2} \log(2\pi e \sum_{t=1}^{T} \sigma_t^2)$, and generalize this to BALD, which is reformulated as $\text{BALD}(\boldsymbol{x}, \boldsymbol{\sigma^2}) = \text{ENTR}(\boldsymbol{x}, \boldsymbol{\sigma^2}) - \frac{1}{T} \sum_{t=1}^{T} [\frac{1}{2} \log(2\pi e \sigma_t^2)]$. Note that these entropy-based formulations exclude predicted values. To get the best of both worlds, acquisition functions that bring together both predicted values and standard deviations could also be constructed.

