# OpenReview forum: "Active Learning for Optimal Minimization of Experimental Characterization Uncertainty"
_NeurIPS.cc/2024/Workshop/BDU — NeurIPS BDU Workshop 2024 Poster_

### Official Review · Reviewer_6kKT · 2024-09-26
**Review of "Active Learning for Optimal Minimization of Experimental Characterization Uncertainty"**

**Rating:** 7
**Confidence:** 3

**Review:**

The paper presents an active learning framework to achieve a cumulative decrease in the uncertainty of key outcome statistics in experimental settings with noisy data. The novelty is that the target for the decrease in uncertainty is a key outcome statistic and not an overall measurement of uncertainty. The authors achieved active learning by selecting regions for measurement without iteratively retraining a model.

Other advantages:
1. The paper is well-written (I could not find typos), and the ideas are mostly clearly presented. However, I think some further discussion is needed regarding the practical benefits of the approach (see major comments)
2. Novel perspective:  The paper challenges the common assumption in active learning that minimizing measurement uncertainty is optimal. Instead, it focuses on minimizing outcome (classification) uncertainty, which is more relevant in many experimental settings.
3. Practical application: The framework is demonstrated on rotational spectroscopy, showing potential for accelerating chemical characterization tasks. This real-world application adds significant value to the paper.

Comments regarding cons and needed improvements:
1. Computational complexity:  The paper does not thoroughly discuss the computational requirements of the method, which could be a concern for real-time or large-scale applications. More discussion on the computational requirements and potential scalability issues would be valuable.
2. It is unclear to me whether the main practical value of the approach is either a) the bottleneck is collecting new data or b) the bottleneck is training the model. The approach requires a pre-trained model, which I presume needs to be trained on the same data (at least, this is the case on the MNIST application where all the images are used). Thus, obtaining a good model will require training on many data; thus, the value is not in point a). It seems, then, that the main value relies on point b), and in that case, I ask what improvements this approach gives compared to the final pre-trained model that could not be obtained by, for instance, training the model for more epochs or with more data. More discussion on this is needed in the paper to clarify the benefits. For instance, it will be nice to see the loss function curve of the training of the pre-trained model preceding the loss function of the AL to check the overall benefit.
 3. Applicability: From my understanding, a dedicated workflow will need to be constructed for each type of task (regression, classification, etc.) and data type, which hinders the approach's broad applicability. The workflow for the two applications applies only to two specific tasks and data types. The approach's applicability would greatly increase by developing a more generic workflow that is more agnostic to the task and data type.
4. The authors could consider some theoretical bounds or guarantees on the performance of their method, at least in a simpler setting and model, such as linear regression.



Conclusion:
The paper's approach of using a fixed pre-trained model and focusing on optimal measurement selection is interesting and well-motivated. This is particularly relevant in experimental settings where retraining models may be impractical or when collecting experimental data is expensive. The authors have demonstrated good scientific thinking by challenging common assumptions and providing a new perspective on active learning for experimental characterization. The discussion of how the method can be adapted for Poisson noise models and regression tasks (Appendix C) extends the applicability of the approach. However, these extensions were not experimentally validated.  My main concert relies on a better discussion and quantification of the benefits, as mentioned in the major comment number two. While there are some areas for improvement, such as broader experimental validation and more theoretical analysis, the paper's strengths outweigh its weaknesses. Overall, I recommend acceptance.

---

### Official Review · Reviewer_zKGc · 2024-09-26
**Motivation is not clear, lacks details on method or key idea**

**Rating:** 4
**Confidence:** 3

**Review:**

The paper proposes an active learning method by combining multiple existing approaches.

-- However, it lacks a clear discussion of the key innovations or any details that differentiate this method from others in the field. For example, it is hard to understand the effect of patch selection loop and how this lead to benefits.

-- Additionally, the claim that standard active learning methods do not account for uncertainty in the outcome is unclear. Methods like Upper Confidence Bound (UCB) explicitly capture the objective value and incorporate uncertainty in their decision-making process.

-- The authors fail to specify which model is used in the paper, which limits the reader's ability to fully understand the proposed approach.

---

### Decision · Program_Chairs · 2024-10-09

Accept (Poster)